# Distribution of Causative Microorganisms in Diabetic Foot Infections: A Ten-Year Retrospective Study in a Tertiary Care Hospital in Central Malaysia

**DOI:** 10.3390/antibiotics12040687

**Published:** 2023-03-31

**Authors:** Parichehr Hadi, Sanjiv Rampal, Vasantha Kumari Neela, Manraj Singh Cheema, Sandeep Singh Sarawan Singh, Eng Kee Tan, Ajantha Sinniah

**Affiliations:** 1Department of Orthopaedic and Traumatology, Faculty of Medicine and Health Sciences, University Putra Malaysia, Serdang 43400, Malaysia; 2Department of Medical Microbiology, Faculty of Medicine and Health Sciences, University Putra Malaysia, Serdang 43400, Malaysia; 3Department of Biomedical Sciences, Faculty of Medicine and Health Sciences, University Putra Malaysia, Serdang 43400, Malaysia; 4Orthopaedic Department, Hospital Ampang, Jalan Mewah Utara, Pandan Mewah, Ampang 68000, Malaysia; 5Department of Pharmacology, Faculty of Medicine, University Malaya, Kuala Lumpur 50603, Malaysia

**Keywords:** diabetic foot, infection, antibiotics, microorganisms, amputation

## Abstract

Diabetes mellitus is a global pandemic, especially in Southeast Asia. Diabetic foot infection (DFI) is a common complication of this condition and causes significant morbidity and mortality in those affected. There is a lack of locally published data on the types of microorganisms and empirical antibiotics being prescribed. This paper highlights the importance of local microorganism culture and antibiotic prescription trends among diabetic foot patients in a tertiary care hospital in central Malaysia. This is a retrospective, cross-sectional study of data taken from January 2010 to December 2019 among 434 patients admitted with diabetic foot infections (DFIs) using the Wagner classification. Patients between the ages of 58 and 68 years old had the highest rate of infection. *Pseudomonas Aeruginosa*, *Proteus* spp., and *Proteus mirabilis* appeared to be the most isolated Gram-negative microorganisms, and *Staphylococcus aureus*, *Streptococcus agalactiae*, and MRSA appeared to be the most common Gram-positive microorganisms. The most common empirical antibiotics prescribed were ampicillin/sulbactam, followed by ciprofloxacin and ceftazidime, and the most common therapeutic antibiotics prescribed were ampicillin/sulbactam, ciprofloxacin, and cefuroxime. This study could be immensely pertinent in facilitating future empirical therapy guidelines for treating diabetic foot infections.

## 1. Introduction

Infected diabetic foot ulcers develop diabetic foot infections (DFIs), which have a high morbidity and fatality rate. Diabetic foot infections are accompanied by a decrease in protective sensitivity (peripheral neuropathy), changed foot architecture, and trauma [1,2]. One-third of diabetic individuals will get a diabetic foot ulcer, and roughly half of these ulcers will become infected. In diabetic patients, foot ulcers are the most common cause of major complications and hospitalization, considerably increasing the disease’s expenses [3,4,5].

Wagner classification is one of the most frequently used and internationally accepted grading systems for diabetic foot ulcers that are used to determine ulcer depth [6,7]. The Wagner classification categorizes tissue injury grade and depth into five stages or levels. The main pathogens in the superficial stages (Wagner 1 and 2) are aerobic bacteria (*Staphylococcus* spp., *Streptococcus* spp., and *Enterobacteriaceae*). Anaerobic flora might also be present in the more severe and deeper stages (Wagner 3, 4, 5) [8]. According to the WHO, the top 10 Asian nations with the highest prevalence of diabetes mellitus (DM) are India, Japan, China, Korea, Bangladesh, Thailand, Indonesia, the Philippines, and Malaysia. Malaysia is expected to experience a remarkable and disturbing growth to 5 million people with DM by 2030, from an estimated 2.5 million in 2018 [9]. Diabetic foot infection (DFI) is a soft tissue or bone infection classified as mild, moderate, or severe. The DFI classification would assist in identifying those patients who should be hospitalized and may require surgical procedures, such as amputation [10].

A variety of infecting microorganisms commonly cause diabetic foot infections. Some common infections that lead to progressive tissue infection are *Staphylococcus aureus*, *Streptococcus Pyogenes*, *Staphylococcus epidermidis*, *Escherichia coli*, *Pseudomonas aeruginosa*, *Klebsiella pneumoniae*, *Acinetobacter* spp., *Proteus* spp., and *Enterococcus* spp. [11].

Gram-positive bacteria, such as *Staphylococcus aureus*, are the most common pathogens associated with mild and moderate infections. Severe or chronic infections are often polymicrobial. For patients with a more severe infection, including sepsis or skin or soft tissue infections, empirical antibiotic therapy is the first line of defense. In contrast, oral antibiotic combination therapy is usually recommended to decrease the likelihood of target organism resistance; drug–drug interactions may limit combinations [12]. Insufficient early therapy may cause an infection to develop, resulting in the patient needing hospitalization and amputation [13]. Hence, treatments aimed at recognizing causative pathogens can vastly improve the result [14].

This paper has determined the common microorganisms among diabetic foot infection patients at a tertiary care hospital in central Malaysia and the prescription antibiotics among them. The significance of this paper will help patients take specific antibiotics and then help reduce the morbidity and mortality rate among them.

## 2. Results

Figure 1 illustrates the distribution of the population from 2010 to 2019. Malay people had significantly higher percentages in their population compared with other races per year.

Table 1 illustrates the distribution of DFI patients by race. Malay patients had the highest percentage among the other ethnicities at 62.0%, while the Indian population had the lowest rate of DFI at 14.7% compared with other ethnic groups.

Table 2 illustrates the distribution of diabetic foot infection patients by age. Patients between the ages of 58 and 68 had the highest infection rate at 35.3%, followed by those aged 69 and over at 30.0%. Overall, the highest rate of DFIs was among those aged 58–68, with 35.3%, followed by patients aged 69 and above with 30.0%, and 23.5% of patients with DFIs were between the ages of 47–57.

Table 3 illustrates the distribution of DFI patients by gender. A higher percentage of males (62.9%) was identified than females (37.1%).

Table 4 shows the distribution of patients based on the Wagner classification. Wagner grades 4 and 5 had the highest percentages among diabetic foot infection patients, with 40.4% and 24.1%, respectively. Moreover, grade 1 had the lowest rate among patients, with 2.1%.

Table 5 illustrates the distribution of Gram-positive and Gram-negative bacteria among DFI patients. The Gram-negative bacteria were determined in 129 (29.7%) DFI patients, and, on the other hand, 84 (19.4%) patients were diagnosed with infections by Gram-positive bacteria. Interestingly, mixed growth infection had a substantial rate among DFI patients, with 80 (18.4%). The data show that *Pseudomonas aeruginosa* (9.4%) was the most common causative Gram-negative microorganism, followed by *Proteus* spp. (4.6%), *Proteus mirabilis* (2.5%), and *Escherichia coli* (2.5%).

The most prevalent Gram-positive bacteria were *Staphylococcus aureus* (6.0%), followed by *Streptococcus agalactiae* (4.1%) and *MRSA* (2.8%).

Table 6 illustrates the antibiotics prescribed to patients with DFIs. The most common empirical antibiotics prescribed were ampicillin/sulbactam at 61.1%, followed by ciprofloxacin at 6.9% and ceftazidime at 6.9%. The most common therapeutic antibiotics prescribed to DFI patients in this study included ampicillin/sulbactam at 31.1%, followed by ciprofloxacin at 17.6% and cefuroxime at 12.2%.

Table 7 illustrates the types of amputations among DFI patients. All patients had amputations, whether minor or major, but mostly had minor amputations (70.7%).

Table 8 illustrates the associations between gender and types of amputations in DFI patients. The statistics show that 37.8% of females had major amputations as compared with 62.2% of males with major amputations. In total, 63.2% of males had minor amputations compared with 36.8% of females with minor amputations. There were no statistically significant differences between gender and types of amputations with *p*-values greater than 0.05. Therefore, there is no association between gender and amputation.

Table 9 shows the associations between types of amputations and empirical antibiotics used among patients with diabetic foot infections. There were no statistically significant differences between the empirical antibiotics used and the types of amputation among diabetic foot infections patients with *p*-values of more than 0.05.

Table 10 shows the associations between the most common microorganisms and types of amputations, whether major or minor. The *p*-values for *Pseudomonas aeruginosa*, *Proteus* spp., *Escherichia coli*, MRSA and *Enterococcus* spp. were less than 0.05; therefore, there was a significant association between them and amputation among diabetic foot infection patients. *Pseudomonas aeruginosa*, *Staphylococcus aureus*, and *Proteus* spp. were the most predominant bacteria in minor amputations, while *Escherichia coli* and *Enterococcus* spp. had a higher percentage in major amputations. The notable point is that *Staphylococcus aureus* had considerable prevalence in both major and minor amputations. The prevalence of Gram-negative bacteria in both major and minor amputations was higher than that of Gram-positive bacteria. Among the Gram-positive bacteria, *Staphylococcus aureus* and *Streptococcus agalactiae* had a higher amputation rate, while among the Gram-negative bacteria, *Pseudomonas aeruginosa* and *Proteus* spp. were the most common microorganisms that contributed to amputation.

## 3. Discussion

Diabetes mellitus (DM) is a major health issue prevalent around the world [15]. With its rising incidence, complications and hospitalizations related to DM are also on the rise [16]. According to a report conducted across many centers in China, patients suffering from diabetic foot disease had a 67.9% increased risk of developing foot infections. It has been claimed that the prevalence of infections in diabetes patients in the Middle East is greater than 40% [17]. Moreover, people with diabetes have approximately a 25% probability of getting a foot ulcer in their lifespan [18].

According to the results of two recent National Health and Morbidity Surveys, the prevalence of diabetes mellitus among Malaysian people increased significantly from 8.3% in 1996 to 14.9% in 2006 (an increase of 80% over ten years), and approximately one-third (36%) of patients were still undiagnosed [19].

This study found the greatest infection rate among Malays, followed by Chinese and Indians and those aged between 58 and 68. Most individuals were male.

A similar finding was reported by several studies [20,21,22,23,24].

According to the Wagner classification, 40.4% of our cases had Wagner grade 4. A similar result was reported in a study conducted in India [25]. A diabetic foot lesion of Wagner grade 4 indicates localized gangrene changes, which are symptomatic of severe necrosis and inadequate circulation in the local tissue [26].

The present study reports a higher prevalence of Gram-negative bacteria (29.7%), contrary to Gram-positive bacteria (19.4%). This finding is supported by previous research conducted in Malaysia and Bermuda [6,27,28]. In contrast, studies from Western countries demonstrate that most of the microorganisms identified with DFI are Gram-positive [29], while Gram-negative bacteria are prevalent in DFI in developing countries in the Southeast Asian (SEA) regions. Nine out of ten studies in SEA regions reported that the proportion of Gram-negative pathogens in DFI was higher [30,31]. Environmental factors, such as sanitary habits, e.g., the use of water for perianal washing (ablution) after defecation, can frequently cause the contamination of hands with fecal flora containing numerous Gram-negative bacteria and are predicted to account for the increasing Gram-negative infections in the developing world compared with Western countries [29,32].

Diabetic foot infections are frequently complicated and can be multi-microbial [33]. While mild DFIs are mainly monomicrobial, severe and moderate DFIs are mainly polymicrobial [29].

Generally, diabetic foot infections are often polymicrobial [34]. Previous research has shown a high incidence (80% to 87.2%) of polymicrobial infections in DFIs [35,36]. Nonetheless, our finding shows that only 18.4% of the pathogens were polymicrobial. Similarly, one study conducted in Egypt reported that 52% of microorganisms were monomicrobial [37]. The low severity of most infections and the lower pathogenicity of isolated bacteria in our studies may account for the low incidence of polymicrobial infections. This agrees with local research conducted in Malaysia [38].

In our study, the most common Gram-negative microorganisms isolated were *Pseudomonas aeruginosa* (9.4%), followed by *Proteus* spp. (4.6%), *Proteus mirabilis* (2.5%), and *Escherichia coli* (2.5%). The observed prevalence of *Pseudomonas aeruginosa* is not unexpected, as it is aligned with previous research by Al-Hamead Hefni et al. that demonstrated a 19.4% rate among all isolates [37]. Moreover, studies in India and Kuwait also found high percentages of *Pseudomonas aeruginosa*, with 29.8% and 17.5%, respectively [34,39]. This finding could be partially explained by the postulated opportunistic nature of *Pseudomonas aeruginosa* in acute and nonhealing lesions [32].

The most predominant Gram-positive bacteria included *Staphylococcus aureus*, (6.0%), *Streptococcus agalactiae* (4.1%), *MRSA* (2.8%), and *Enterococcus* spp. (2.5%). Several studies found that *Staphylococcus aureus* is the most prevalent pathogen in diabetic foot infections [27,40,41,42]. However, in the present study, it was the second-most-common isolated pathogen after *Pseudomonas aeruginosa.* Similar results have been observed in other studies conducted in Malaysia, India, and Turkey [32,39,43].

In the beginning, the selection of antibiotics needs to be made on an empirical basis; however, after the culture reports are known, the choice must be particular and narrowed down [44]. According to one study conducted in Malaysia, ampicillin + sulbactam was the most preferred empirical antibiotic, followed by clindamycin and ceftazidime. The antibiotics provided reduce the chance of amputation and, thus, improve the prognosis of the infection [45]. A similar result was found in our study; ampicillin + sulbactam was the most common antibiotic prescribed among diabetic foot infection patients. The ampicillin/sulbactam is often reserved for patients in which bacteria have developed resistance to other antibiotics, such as cloxacillin and penicillin [45]. Furthermore, for limb-threatening diabetic foot infections, ampicillin/sulbactam is more cost-effective than imipenem/cilastatin [6,46]. The local trends for using these antibiotics are based on the local CPG (clinical practice guidelines) for diabetic foot management [47].

Amputation is the removal of a limb that is no longer viable and should be considered for individuals with infections that cannot be controlled or wounds that do not heal [48]. Amputation risk factors in people with DFI include age, gender (male), comorbidities, or the consequences of diabetes [49]. In a few studies, the authors reported on particular pathogens to evaluate prediction variables for limb loss; for example, they reported that DFIs infected with *Staphylococcus aureus* were a predicted factor for limb loss [28,50]. The present study shows that *Staphylococcus aureus* had a considerable rate in both minor and major amputations.

Therefore, identifying the microbes of diabetic foot infection is vital, as a lack of knowledge could cause the improper management of this condition and could exacerbate increased infections, increased hospital admissions, and, eventually, amputations [28]. Thus, antibiotic treatment and wound control are critical for limb preservation [42].

Based on the study’s currently accessible data, this study may help clinicians to choose the best antibiotic for the cultured microorganisms. The current study has limitations, including poor recordkeeping and the loss of information due to the digitalization of patients’ medical records. In our research, there was a high percentage of no growth reported. We postulate that bacteria cannot grow in the laboratory in improper environmental conditions such as due to nitrogen sources, temperature, osmotic conditions, and pH [51]. Despite this limitation, this analysis is crucial because it offers information on the exposure of patients to the bacteria that cause DFIs. This information is vital for developing hospital policies that allow for the appropriate treatment of DFIs. This paper is considered novel in Malaysia because there is a lack of studies regarding prescription trends, the total number of patients in nine years, and causative microorganisms, which were most Gram-negative.

## 4. Materials and Methods

This is a cross-sectional study using retrospective data taken from January 2010 to December 2019 among 434 patients with DFIs in a tertiary care hospital in central Malaysia. The retrospective data of the DFI patients were collected from their electronic medical records (EMRs), and to collect all the required data, a proforma was used. The data collected from patients’ records included race, age, gender, Wagner classification, microorganisms, antibiotics prescribed, and amputation. In this study, the types of infections included superficial, skin, soft tissue, and osteomyelitis bone. All adults who were admitted to the hospital with DFI as a primary diagnosis were included. The criteria of diagnosis depended on clinical examinations that included the size and depth of the ulcer according to Wagner classification; the presence of elevated inflammatory markers that would include ESR, CRP, and increased total WBC; and the presence of osteomyelitis changes in a radiograph of the foot. We used Wagner because it is easier for clinicians to classify the severity of diabetic foot infections.

Exclusion criteria included in this study were admission for other causes of foot infections, such as dermatology and dry gangrene of the foot.

All the data were analyzed using SPSS (Statistical Software Package for Social Science) for Microsoft Windows, version 27. Descriptive statistics were used to describe the characteristics of patients, antibiotics prescribed, and the causative microorganisms identified. These are presented as percentages and frequencies and used to describe the variables’ distributions. Furthermore, categorical data analysis (Pearson Chi-square test or Fisher’s exact test) was used. A *p*-value < 0.05 was considered statistically significant. The study was approved by the Medical Research and Ethics Committee, Ministry of Health, Malaysia, NMRR ID-22-00250-PB8 (IIR).

## 5. Conclusions

In conclusion, the highest rate of diabetic foot infection was identified among patients between the ages of 58 and 68. Most patients had Wagner grades 4 and 5. Gram-negative bacteria had higher percentages (29.7%) than Gram-positive (19.4%) bacteria. The most common Gram-negative bacterium was *Pseudomonas aeruginosa*, followed by *Proteus* spp. and *Proteus mirabilis*. In contrast, the most common Gram-positive bacterium was *Staphylococcus aureus*, followed by *Streptococcus agalactiae* and *MRSA.* The most predominant empirical antibiotic prescribed was ampicillin/sulbactam, followed by ciprofloxacin and ceftazidime, while the most common therapeutic antibiotics included ampicillin/sulbactam, ciprofloxacin, and cefuroxime. The prevalence of Gram-negative bacteria in both major and minor amputations was higher than that of Gram-positive bacteria.

## Figures and Tables

**Figure 1 antibiotics-12-00687-f001:**
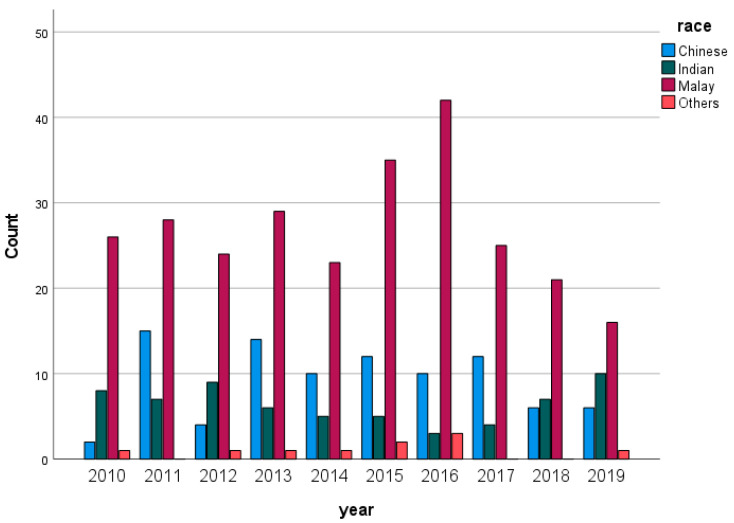
Distribution of populations with diabetic foot infections from 2010 to 2019.

**Table 1 antibiotics-12-00687-t001:** Distribution of DFI patients by race.

Race	Number of Patients (n)	Percentage (%)
Malay	269	62.0%
Chinese	91	21.0%
Indian	64	14.7%
Others	10	2.3%
Total	434	100.0%

Others: Indonesian and Burmese.

**Table 2 antibiotics-12-00687-t002:** Distribution of DFI patients by age.

Age Categories	Number of Patients (n)	Percentages (%)
25–35	7	1.6%
36–46	42	9.7%
47–57	102	23.5%
58–68	153	35.3%
Age 69 and over	130	30.0%
Total	434	100.0%

**Table 3 antibiotics-12-00687-t003:** Distribution of DFI patients by gender.

Gender	Number of Patients (n)	Percentage (%)
Male	273	62.9%
Female	161	37.1%
Total	434	100.0%

**Table 4 antibiotics-12-00687-t004:** Distribution of patients regarding Wagner classification.

Wagner Classification	n, (%) Patients	Clinical Outcome *
Grade 1	9 (2.1)	Superficial ulcer
Grade 2	77 (18.0)	Deep ulcers involving capsule, tendon, bone
Grade 3	66 (15.4)	Deep ulcers with abscess osteomyelitis or joint sepsis
Grade 4	173 (40.4)	Localized gangrene
Grade 5	103 (24.1)	Whole-foot gangrene
Total	428 (100.0)	
Missing data	6 (1.4)	

* Source: adapted from [15].

**Table 5 antibiotics-12-00687-t005:** Distribution of Gram-positive and Gram-negative bacteria.

Microorganisms	Number of Microorganisms (n)	Percentages (%)
Gram-negative bacteria	129	29.7%
*Pseudomonas aeruginosa*	41	9.4
*Proteus* spp.	20	4.6
*Proteus mirabilis*	11	2.5
*Escherichia coli*	11	2.5
*Morganella morganii*	9	2.1
*Acinetobacter* spp.	8	1.8
*Klebsiella* spp.	7	1.6
*Aeromonas hydrophila*	4	0.9
*Stenotrophomonas maltophilia*	4	0.9
*Klebsiella* spp. (*ESBL producer*)	4	0.9
*Klebsiella pneumoniae*	3	0.7
*Escherichia coli* (*ESBL producer*)	1	0.2
*Bacteroides fragilis*	1	0.2
*Enterobacter cloacae*	1	0.2
*Enterobacter* spp.	1	0.2
*Klebsiella oxytoca*	1	0.2
*Serratia liquefaciens*	1	0.2
*Serratia marcescens*	1	0.2
Gram-positive bacteria	84	19.4%
*Staphylococcus aureus*	26	6.0
*Streptococcus agalactiae*	18	4.1
*MRSA*	12	2.8
*Enterococcus* spp.	11	2.5
*Streptococcus* spp.	8	1.8
*Coagulase-negative Staphylococcus species*	7	1.6
*Bacillus* spp.	1	0.2
*Corynebacterium* spp.	1	0.2
Mixed growth	38	8.7%
*Klebsiella pneumoniae*, *Streptococcus agalactiae*	2	0.5
*Staphylococcus aureus*, *bacillus* spp.	2	0.5
*E. coli*, *Acinetobacter*, *Enterobacter*	1	0.2
*E. coli*, *Klebsiella*	1	0.2
*E. coli*, *Proteus mirabilis*	1	0.2
*Enterobacter cloacae*, *Enterococcus* spp.	1	0.2
*Enterobacter* spp., *pseudomonas aeruginosa*	1	0.2
*Enterobacter*, *Klebsiella* spp.	1	0.2
*Enterococcus* spp., *Coagulase-negative Staphylococcus species*	1	0.2
*Klebsiella pneumonia*, *Enterococcus*	1	0.2
*Klebsiella*, *Morganella Morganii*	1	0.2
*Klebsiella ozaenae*, *Staphylococcus aureus*	1	0.2
*Klebsiella pneumoniae*, *Pseudomonas* spp.	1	0.2
*Klebsiella*, *E. coli*, *Pseudomonas* spp.	1	0.2
*Klebsiella*, *pseudomonas aeruginosa*	1	0.2
*Klebsiella*, *Pseudomonas* spp.	1	0.2
*Morganella morganii*, *Pseudomonas aeruginosa*	1	0.2
*Morganella morganii*, *Serratia*	1	0.2
*Proteus Mirabilis*, *Citrobacter*	1	0.2
*Proteus mirabilis*, *staphylococcus aureus*	1	0.2
*Proteus* spp., *Streptococcus agalactiae*	1	0.2
*Proteus* spp., *klebsiella*, *staphylococcus aureus*	1	0.2
*Pseudomonas aeruginosa*, *Enterococcus* spp.	1	0.2
*Pseudomonas aeruginosa*, *Klebsiella*, *Enterobacter*	1	0.2
*Pseudomonas aeruginosa*, *Proteus* spp.	1	0.2
*Pseudomonas aeruginosa*, *Streptococcus agalactiae*	1	0.2
*Pseudomonas*, *Klebsiella*, *Proteus* spp.	1	0.2
*Serratia liquefaciens*, *Escherichia coli*, *Streptococcus agalactiae*	1	0.2
*staphylococcus aureus*, *Streptococcus* spp., *burkhoideria* spp., *proteus* spp.	1	0.2
*Staphylococcus aureus*, *Enterococcus*	1	0.2
*Staphylococcus aureus*, *Streptococcus agalactiae*	1	0.2
*Staphylococcus*, *streptococcus viridans*, *bacillus* spp.	1	0.2
*Staphylococcus* spp., *E. coli*, *Proteus* spp., *Klebsiella* spp.	1	0.2
*Staphylococcus*, *Pseudomonas*, *Proteus* spp.	1	0.2
*Staphylococcus*, *streptococcus viridans*, *streptococcus group F*, *bacillus* spp.	1	0.2
*Streptococcus*, *Enterococcus*	1	0.2
Heavily mixed growth	6	1.4
Mixed growth	36	8.3
NG	141	32.5
**Total**	**434**	**100.0**

MRSA: Methicillin-resistant Staphylococcus aureus. NG: No growth.

**Table 6 antibiotics-12-00687-t006:** Antibiotics prescribed for DFI patients.

Type of Therapy	Number of Patients (n)	Percentages (%)
Empirical Antibiotics	144	100.0%
Ampicillin/sulbactam	88	61.1%
Ciprofloxacin	10	6.9%
Ceftazidime	10	6.9%
Ceftazidime, clindamycin	6	4.2%
Other antibiotics or antibiotic combinations	30	20.8%
Therapeutic Antibiotics	74	100.0%
Ampicillin/sulbactam	23	31.1%
Ciprofloxacin	13	17.6%
Cefuroxime	9	12.2%
Ceftazidime	6	8.1%
Other antibiotics or antibiotic combinations	23	31.1%
Not Specified Antibiotics	202	100.0%
Ampicillin/sulbactam	111	53.6%
Cefuroxime	18	8.7%
Ceftazidime	7	3.4%
Ceftazidime, clindamycin	10	4.8%
Other antibiotics or antibiotic combinations	56	27.1%
Missing data	14	

Empirical antibiotics: They were administered based on clinical diagnosis. Therapeutic antibiotics: Causative agents confirmed by laboratorians, and then antibiotics were administered for certain targets. Not specified antibiotics: In the data, it was not specified whether the antibiotics were empirical or therapeutic.

**Table 7 antibiotics-12-00687-t007:** Types of amputations.

Amputation	Number of Amputations (n)	Percentages (%)
Minor	307	70.7%
Major	127	29.3%
Total	434	100.0%

**Table 8 antibiotics-12-00687-t008:** Association between gender and types of amputations.

Gender	Major Amputation	Minor Amputation	X^2^ Value	*p*-Value
Male	79 (62.2%)	194 (63.2%)	0.03	0.84
Female	48 (37.8%)	113 (36.8%)		

**Table 9 antibiotics-12-00687-t009:** Association between amputations and empirical antibiotics used.

Empirical Antibiotics	Major Amputation	Minor Amputation	X^2^ Value	*p*-Value
Ampicillin/sulbactam	13 (14.8%)	75 (85.2%)	3.07	0.08
Ciprofloxacin	0 (0.0%)	10 (100.0%)	1.34	0.60
Ceftazidime	1 (10.0%)	9 (90.0%)	0.01	1.000
Ceftazidime, clindamycin	0 (0.0%)	6 (100.0%)	0.78	1.000
Other antibiotics	2 (6.7%)	28 (93.3%)	0.75	0.52

**Table 10 antibiotics-12-00687-t010:** Associations between amputations and the most common Gram-negative and Gram-positive microorganisms.

Microorganisms	Major Amputation	Minor Amputation	X^2^ Value	*p*-Value
*Pseudomonas aeruginosa*	2 (4.9%)	39 (95.1%)	13.00	0.001
*Proteus* spp.	1 (5.0%)	19 (95.0%)	5.96	0.01
*Proteus mirabilis*	2 (18.2%)	9 (81.8%)	0.66	0.52
*Escherichia coli*	11 (100.0%)	0 (0.0%)	27.28	0.001
*Staphylococcus aureus*	7 (26.9%)	19 (73.1%)	0.07	0.78
*Streptococcus agalactiae*	2 (11.1%)	16 (88.9%)	2.98	0.08
*MRSA*	0 (0.0%)	12 (100.0%)	5.10	0.02
*Enterococcus* spp.	10 (90.9%)	1 (9.1%)	20.72	0.001

## Data Availability

Not applicable.

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
