# Peer review of "Distribution of Causative Microorganisms in Diabetic Foot Infections: A Ten-Year Retrospective Study in a Tertiary Care Hospital in Central Malaysia"

_antibiotics, 2023, doi:10.3390/antibiotics12040687_

Round 1

Reviewer 1 Report

1- Matherials and methods: this part could be anticipated just after introduction.

2- in nine years microbiological population could change, it is important to check if there is difference between 2010 and 2019

3- gravity of infections are not described. I suggest to describe level of infection at presentation to better correlate outcomes

4- in the article (introduction and discussion) there is a long discussion about correlation between gravity of infection and presence of mono-polibacterial in samples but these data are not reported

5- resistances to antibiotics are not reported 

6- explain the large number of no growth samples

Reviewer 2 Report

Thanks for giving me the opportunity to review those paper. I think some amendments is needing for improving the manuscript 

1.- Please move the Material and Methods section before the results and discussion section

2.- Describe in Material and Methods section the criteria of diagnosis of diabetic foot infections 

3.-It is not clear if all patients including in the study suffered from serious infection. Do you include patients with moderate and mild infection as well?

4.- What kind of infections were included?: soft tissue infections or bone infection or both. Please describe in M&M 

5.- What was the vascular status of the patients?. Please describe this data and its potential impact in the patient outcomes

6.- Table 2. Why so you split the patients in this age ranges?

7.- The distribution of the bacteria according with gram is not clear. Please review the percentage.

8.- Please don't use comercial brand of ATB within the manuscript such us Fortum or Unsayn. Use the principle ATB description

9- In the table describing use of ATB you describe empirical, therapeutic and not specific. Please clarify what does it mean?

10-. The association between amputation and ATB is not clear. Please discuss at the discussion section why this bacteria could be associated with either major or minor amputation 

11.- Discussion should be rewrite and according with the order of the results presentation. 

12.- Please discuss why in your hospital you use this empirical ATB. Is it coming from local protocol?

13.- Please discuss the effect of any ATB used in your hospital 

14.- Please limit your conclusion to the results of your study 

Round 2

Reviewer 2 Report

Dear Authors, 

Thanks for answered my comments and addressed some of my recommendations. However there are still some pending issues that must be addressed:

1.- You mentioned that you used Wagner classification as a method of diagnosis of infection and this statement in not correct. Wagner is using for clasifying the severity of the ulcer, but it is not related with the presence of infection, especially in stage 1 and 2. Stage 3 is supposed associated with OM but not always a bone exposed is a bone infected, or at least 100%. Wagner 4 and 5 are gangrene, but again not related with infection, because gangrene can be caused by ischemia and not necessary in all cases by infection. Wagner is not recognized as a severity infection classification, International the most used are IDSA and PEDIS, but regardless this argument you can not be use Wagner as method of diagnosis, because Wagner clarify the ulcer after ulcer assessment and diagnostic, not the opposite. Please amend this main concern in your study.

- Authors mentiones that terry addressed the dicussion section, but it there didn't it, because the only thing that they did was adding some additional information, but not develop a real discussion, that MUST discuss the results of the study.....it is not an space for philosophy or own ideas debate. You should start highlight the main results of your study, them you should compare these results with previos similar studies, them you should give your opinion about why you got these results, finally you have to describe your strengths and your limitations. 
